# Insulin-Like Growth Factor 2 mRNA-Binding Protein 1 (IGF2BP1) Is a Prognostic Biomarker and Associated with Chemotherapy Responsiveness in Colorectal Cancer

**DOI:** 10.3390/ijms22136940

**Published:** 2021-06-28

**Authors:** Hung-Ming Chen, Chun-Chi Lin, Wei-Shone Chen, Jeng-Kai Jiang, Shung-Haur Yang, Shih-Ching Chang, Ching-Liang Ho, Chung-Chi Yang, Shih-Ching Huang, Yee Chao, Tsai-Tsen Liao, Wei-Lun Hwang, Hao-Wei Teng

**Affiliations:** 1Department of Medicine, Division of Hematology and Oncology, Taoyuan Armed Forces General Hospital, Taoyuan 325, Taiwan; Shedy1007@gmail.com; 2Department of Medicine, Division of Hematology and Oncology, Tri-Service General Hospital, National Defense Medical Center, Taipei 114, Taiwan; Charileho22623@gmail.com; 3Faculty of Medicine, School of Medicine, National Yang Ming Chiao Tung University, Taipei 112, Taiwan; cclin15@vghtpe.gov.tw (C.-C.L.); wschen@vghtpe.gov.tw (W.-S.C.); jkjiang@vghtpe.gov.tw (J.-K.J.); yangsh@vghtpe.gov.tw (S.-H.Y.); changsc@vghtpe.gov.tw (S.-C.C.); ychao@vghtpe.gov.tw (Y.C.); 4Department of Surgery, Division of Colon and Rectal Surgery, Taipei Veterans General Hospital, Taipei 112, Taiwan; 5Department of Surgery, National Yang Ming Chiao Tung University Hospital, Yilan 260, Taiwan; 6Department of Medicine, Division of General Medicine, Taoyuan Armed Forces General Hospital, Taoyuan 325, Taiwan; t220979@gmail.com; 7Department of Medicine, Division of General Medicine, Tri-Service General Hospital, National Defense Medical Center, Taipei 114, Taiwan; 8Department of Biotechnology and Laboratory Science in Medicine, National Yang Ming Chiao Tung University, Taipei 112, Taiwan; v820707d@yahoo.com.tw; 9Department of Oncology, Division of Medical Oncology, Taipei Veterans General Hospital, Taipei 112, Taiwan; 10Institute of Clinical Medicine, National Yang Ming Chiao Tung University, Taipei 112, Taiwan; 11Graduate Institute of Medical Science, College of Medicine, Taipei Medical University, Taipei 110, Taiwan; liaotsaitsen@tmu.edu.tw; 12Cancer Progression Research Center, National Yang Ming Chiao Tung University, Taipei 112, Taiwan

**Keywords:** *IGF2BP1*, colorectal cancer, prognosis, biomarker

## Abstract

Insulin-like growth factor 2 mRNA-binding protein 1 (IGF2BP1) is an RNA-binding protein and serves as a post-transcriptional fine-tuner regulating the expression of mRNA targets. However, the clinicopathological roles of IGF2BP1 in colorectal cancer (CRC) remains limited. Thus, we aimed to elucidate the clinical significance and biomarker potentials of IGF2BP1 in CRC. A total of 266 specimens from two sets of CRC patients were collected. IGF2BP1 expression was studied by immunohistochemical (IHC) staining. The Kaplan-Meier survival plot and a log-rank test were used for survival analysis. The Cox proportional hazards model was applied to determine the survival impact of IGF2BP1. Public datasets sets from The Cancer Genome Atlas (TCGA) and Human Cancer Metastasis Database (HCMDB), receiver operating characteristic (ROC) plotter, and two CRC cell lines, HCT-116 and DLD-1, were used for validating our findings. We showed that IGF2BP1 was overexpressed in tumor specimens compared to 13 paired normal parts by examining the immunoreactivity of IGF2BP1 (*p* = 0.045). The increased expression of *IGF2BP1* in primary tumor parts was observed regardless of metastatic status (*p* < 0.001) in HCMDB analysis. IGF2BP1 expression was significantly associated with young age (59.6% vs. 46.7%, *p*-value = 0.043) and advanced stage (61.3% vs. 40.0%, *p*-value = 0.001). After controlling for confounding factors, IGF2BP1 remained an independent prognostic factor (HR = 1.705, *p*-value = 0.005). TCGA datasets analysis indicated that high *IGF2BP1* expression showed a lower 5-year survival rate (58% vs. 65%) in CRC patients. The increased expression of *IGF2BP1* in chemotherapy non-responder rectal cancer patients was observed using a ROC plotter. Overexpression of *IGF2BP1* promoted the colony-forming capacity and 5-fluorouracil and etoposide resistance in CRC cells. Here, IGF2BP1 was an independent poor prognostic marker in CRC patients and contributed to aggressive phenotypes in CRC cell lines.

## 1. Introduction

Colorectal cancer (CRC) is one of the most malignant cancers globally. It was estimated that more than 1.8 million new CRC cases and 881,000 CRC-related deaths occurred in 2018 [1,2]. Unfortunately, around one-third of patients with stage I–III CRC finally advanced to metastatic CRC (mCRC). Finally, 30–40% of patients with CRC were stage IV or mCRC. Despite advances in chemotherapy, including 5-fluorouracil (5-FU), irinotecan, oxaliplatin, bevacizumab, cetuximab, regoragenib, lonsurf, BRAF inhibitors, and anti-PD-L1/PD-1 agents, the median overall survival (OS) in mCRC is approximately 30–36 months [3,4,5]. After therapeutic treatment, CRC patients often suffered from drug resistance, which is positively associated with poor prognosis of CRC [4]. Therefore, there is a need to find new druggable targeted genes for treating patients with metastatic CRC.

CRC is a heterogeneous disease with three major pathways operating within CRC carcinogenesis, namely the chromosomal instability (CIN) pathway, which is the most well-characterized, microsatellite instability (MSI), and epigenetic alteration through CpG island methylator phenotype (CIMP) [6,7]. Most of CRC follows the Wnt signaling pathway, Ras pathway, p53 system, and other pathways involved in CIN [7]. In addition, clinically significant biomarkers of CRC are required not only for early detection but also for accurate diagnosis, classification, predication drug efficacy, and prognosis and remark of CRC following treatment.

The insulin-like growth factor (IGF) signaling pathway is critical to the tumorigenesis, angiogenesis, and maintenance of many tissues within the normal cad cancer tissue as well as to the stimulation of cell proliferation and interruption of programmed cell death [8]. The IGF system has two ligands, namely IGF-1 and IGF-2, which exhibit their effects through binding to IGF-1R (primarily), IGF-2R, and the insulin receptor (IR), all belonging to the tyrosine kinase receptor family [9]. One of the interesting druggable components of this pathway is the insulin-like growth factor-2 mRNA-binding protein 1 (IGF2BP1).

IGF2BP1, a member of a conserved family of single-stranded RNA-binding proteins (IGF2BP1, 2, and 3), expresses in a broad range of fetal tissues and more than 16 cancers but only in a limited number of normal adult tissues [10]. IGF2BP1 has an essential role in embryogenesis, tumorigenesis, and chemoresistance via serving as a post-transcriptional regulation by regulating the expression of some essential mRNA targets required for tumor cell growth and proliferation invasion as well as chemotherapy resistance [2,8,9,11,12,13,14,15,16,17,18,19], resulting in poor overall survival and metastasis in various types of cancers [10,20]. Similar to IGF2BP1, IGF2BP3 expression correlates with a poor prognosis and tumor aggressiveness in several cancers including gastrointestinal cancers [21], lung cancer [22], and melanoma [23]. Together with IGF2BP1, IGF2BP3 facilitates invadopodia formation and cancer metastasis by preventing the degradation of the *CD44* mRNA [24]. IGF2BP2, known as an N6-methyladenosine (m6A) reader, regulates cell metabolism and cancer progression by interacting with different RNA species [25,26]. Therefore, IGF2BPs have been regarded as oncogenes and potential druggable targets for cancers. Nevertheless, the clinical impact of IGF2BP1 on CRC remains elusive. In this study, we aimed to elucidate the clinical significance of IGF2BP1 expression in CRC patients and validate the roles of IGF2BP1 in CRC cell lines.

## 2. Results

### 2.1. IGF2BP1 Was Overexpressed in CRC Tumors Comparing to Normal Counterparts

In an attempt to explore the clinical significance of IGF2BP1 in the tumorigenesis and progression of CRC, we collected two sets of CRC specimens and examined the expression level of IGF2BP1 at the protein level by IHC staining. The *IGF2BP1* mRNA expression data from public domains were used for further validation. A total of 13 paired tumor–normal samples was used in evaluating the expression of IGFBP1. The representative IHC images of immunoreactivity strong IGF2BP1 expression in tumors (Figure 1A) and weak IGF2BP1 expression in paired normal parts (Figure 1B) were shown. Grossly, IGF2BP1 was overexpressed in tumor parts comparing to normal specimens (Figure 1C, *p*-value = 0.045). Two patients (one stage II and one stage IV adenocarcinoma patients without receiving any treatment) had low IGF2BP1 H-scores compared to paired adjacent normal tissues (Figure 1C). The increased expression of *IGF2BP1* transcript was also observed in primary tumors without metastasis (Figure 1D, EXP00386) and with metastasis (Figure 1E, EXP00363), but not in EXP00100 datasets analyzed in comparison to corresponding normal colonic tissues by analyzing the HMDCB.

### 2.2. IGF2BP1 Expression Was Associated with Late Stage (III/IV) Status and Poor Overall Survival in CRC Patients

As IGF2BP1 showed increased expression in tumor parts, we next addressed the roles of IGF2BP1 in cancer progression and patient survival. The second set of CRC specimens from a total of 253 patients were collected for analysis. The representative IHC images of weak and strong IGF2BP1 immunoreactivity are shown in Figure 2A (H-score: 10) and Figure 2B (H-score: 270), respectively.

The association between patients’ characteristics and IGF2BP1 expression is listed in Table 1. IGF2BP1 was significantly associated with young age and advanced stage. The strong IGF2BP1 expression rate in patients <70 y/o and ≥70 y/o was 59.6% and 46.7%, respectively (*p*-value = 0.043). The strong IGF2BP1 expression rate in patients with stage III/IV and stage I/II was 61.3% and 40.0%, respectively (*p*-value = 0.001). IGF2BP1 expression was not associated with gender, pathology, location, pathology grade, mucinous component, or lymph-vascular space invasion (LVSI). We found that strong IGF2BP1 immunoreactivity was significantly associated with poor prognosis, compared to weak IGF2BP1 expression (median overall survival (OS) in strong IGF2BP1 expression was 34.6 months and in weak IGF2BP1 expression was not reached. *p*-value = 0.001). Strong IGF2BP1 expression showed a worse overall survival of 253 CRC patients (Figure 2C).

To validate this finding, the expression data of *IGF2BP1* transcript was retrieved from the two TCGA datasets, namely TCGA-COAD and TCGA-READ, analyzed with Human Pathology Atlas. A total of 577 CRC patients were subjected to Kaplan-Meier (KM) survival analysis (death: 115 patients, alive: 462 patients). CRC patients who showed a high expression of *IGF2BP1* transcript under the best expression cutoff (0.13) exhibited poor survival (Figure 2D). The 5-year survival rate in high (*n* = 163) and low (*n* = 414) expression of *IGF2BP1* transcript was 58% and 65%, respectively. This finding partially provided support to our finding. By analyzing the *IGF2BP1* expression in data sets deposited in HDMCB, we found that primary tumors with metastasis showed enhanced expression of *IGF2BP1* transcript compared to those of primary tumors without metastasis (Figure 2E). The *IGF2BP1* expression was even enhanced in liver metastasized tumors (Figure 2F).

### 2.3. IGF2BP1 Was a Cancer Biomarker for Classifying Chemotherapy Responsiveness in Rectal Cancer Patients

Next, the microarray expression data of a total of 107 CRC patients (91 colon cancer and 16 rectal cancer patients) ≥50 y/o without receiving radiotherapy were retrieved from a ROC plotter web tool (http://www.rocplot.org/ (accessed on 1 June 2021) to examine the association of *IGF2BP1* expression and chemotherapy resistance. Three JetSet best affymetrix probes, including 227377_at*, 223689_at*, and 241574_s-at* were annotated with *IGF2BP1* in the ROC plotter. It was found that the expression of *IGF2BP1* (probe: 227377_at*) was not altered in non-responders to any chemotherapy (5-fluorouracil, oxaliplatin, bevacizumab, irinotecan, and capecitabine) compared to 91 responder colon cancer patients (Figure 3A). The expression of *IGF2BP1* (probe identity: 227377_at*) at the strong cutoff (11) was unable to predict chemotherapy responsiveness in this cohort (Figure 3B). Nevertheless, we found that *IGF2BP1* expression (probe: 227377_at*) was higher in non-responders to any chemotherapy (5-fluorouracil, oxaliplatin, bevacizumab, irinotecan, and capecitabine) than in responder rectal cancer patients (Figure 3C). The expression of *IGF2BP1* (probe: 227377_at*) was considered as a top-quality cancer biomarker for classifying chemotherapy responsiveness in this cohort at the strong cutoff (16) (Figure 3D), suggesting that enhanced *IGF2BP1* expression may contribute to therapeutic resistance in CRC subpopulations. Two other JetSet best affymetrix probes (223689_at* and 241574_s_at*) were unable to predict therapeutic responsiveness in neither colon cancer (Appendix A) nor rectal cancer patients (Appendix A).

### 2.4. IGF2BP1 Was an Independent Prognostic Cancer Biomarker in CRC Patients

To delineate the potential prognostic impact of IGF2BP1 on CRC, a Cox proportional hazards model was used. In univariate analysis according to OS, IGFBP1 (HR = 1.860, *p*-value = 0.001), stage (III/VI vs. II/I) (HR = 1.089, *p*-value < 0.001), mucinous component (HR = 1.636, *p*-value = 0.005), and LVSI (HR = 2.465, *p*-value < 0.001) were identified as significant prognostic factors in predicting overall survival (Figure 4A). After controlling for other confounding factors, IGF2BP1 (HR = 1.705, *p*-value = 0.005), stage, mucinous component, and LVSI remained independent prognostic factors (Figure 4B).

### 2.5. Overexpression of IGF2BP1 Increased Colony-Forming Capacity and Conferred Drug Resistance in CRC Cell Lines

As the IGF2BP1-driven malignant phenotypes observed in clinical specimens may have resulted from intercellular interactions between IGF2BP1-expressing CRC cells and neighboring host cells within a tumor microenvironment (TME) or IGF2BP1-experssing CRC cells alone. Here, we established ectopically IGF2BP1-expressing HCT-116 and DLD-1 CRC cells for examining the impact of IGF2BP1 directly on cancer cells. The ectopic expression of IGF2BP1 in HCT-116 and DLD-1 was confirmed by western blotting (Figure 5A). We found that IGF2BP1 overexpression increased colony formation (Figure 5B,C), compared to those of vector control cells. Additionally, we found that overexpression of IGF2BP1 increased drug resistance to 5-FU (Figure 5D) and etoposide (Figure 5E) in both HCT-116 and DLD-1 cells.

## 3. Discussion

Numerous human cancers have been shown to be associated with aberrant IGF signal pathways, including colorectal cancer, breast cancer, lung cancer, pancreatic cancer, melanoma, and childhood cancers, among many others [27]. Numerous in vitro and clinical studies have disclosed that increased IGF-2 activity is implicated in cancer cell proliferation, migration, and invasion [12,13,15,27]. The IGF system has two ligands, namely IGF-1 and IGF-2, and three receptors, namely IGF-1R (primarily), IGF-2R, and IR [7,8,12,15]. IGF2BP1 can interact and process the *IGF2* transcripts, and its dysfunction was associated with tumorigenesis, drug resistance, and diabetes as well as insulin control [8,14,15,28]. These reports indicate that IGF2BP1 is oncogenic.

It has been known that CRC is a frequently lethal disease with heterogeneous genomics and clinical outcomes [29]. The CRC Subtyping Consortium (CRCSC) has tried to assess the presence or absence of core subtype patterns among existing gene expression-based CRC subtyping algorithms [29,30,31]. CRC was classified into four consensus molecular subtypes (CMS) with distinguishing features: CMS1 (MSI Immune, 14%), hypermutated, microsatellite unstable, strong immune activation; CMS2 (Canonical, 37%), epithelial, chromosomally unstable, marked WNT, and MYC signaling activation; CMS3 (Metabolic, 13%), epithelial, evident metabolic dysregulation; and CMS4 (Mesenchymal, 23%), prominent transforming growth factor β activation, stromal invasion, and angiogenesis [29,30,31]. In the CMS3 group, it is interesting that enrichment for many metabolic pathways was disclosed. Among the metabolic pathways, one of the most important signatures is the IGF pathway. The IGF pathway plays an important role in tumorigenesis, angiogenesis, and drug resistance, as well as interrupting programmed cell death [7,8,15,18,19,32,33]. Based on the above, IGF2BP1 is an interesting druggable component of this pathway in colorectal cancer.

Different from its oncogenic role previously reported in colorectal and other cancers [16,17], IGF2BP1 had been reported to have tumor suppression roles. In a mouse model of colitis-associated cancer, following the knockdown of *IGF2BP1* in stromal cells, the mice showed elevated tumor burden [34]. Additionally, IGF2BP1 loss enhanced the levels of HGF, which is produced by stromal fibroblasts and contributes to epithelial cell proliferation and invasive growth of CRC cells by interaction with β-catenin signaling, and conferred resistance to EGFR inhibitors in colon tumor-initiating cells in fibroblasts in vitro, and increased fibroblast cell growth [34,35,36,37], indicating a potentially tumor-suppressive role of IGF2BP1 via modulating HGF in fibroblasts. It is interesting that IGF2BP1 plays different roles in different tissues. In this study, we found that overexpression of IGF2BP1 in colorectal cancer cell lines, namely HCT116 and DLD-1, increased colony formation and increased drug resistance to 5-FU and etoposide. Our findings could be supported by Mongroo et al.’s report [17]. The knockdown of c-Myc in colorectal cancer cell lines increases the expression of mature let-7 miRNA family members and downregulates IGF2BP1. The loss of IGF2BP1 inhibits Cdc34, Lin-28B, and K-Ras, suppresses cancer cell proliferation and anchorage-independent growth, and promotes caspase-mediated cell death. Moreover, overexpression of IGF2BP1 increases c-Myc and K-Ras expression and LIM2405 cell proliferation. They indicated that IGF2BP1, interrelated with c-Myc, acts upstream of K-Ras to promote survival, which was a novel mechanism important in colon cancer pathogenesis [17].

Our study had some limitations. First, we did not have genomics data about *RAS*, *BRAF*, and MSI; thus, we could not clarify the interaction between *IGF2BP1* and these important genomic features. Second, the SPSS visual binning function cutoff level (100) applied in the IHC assay for patient stratification might lead to bias. Third, although the expression of *IGF2BP1* was shown to be associated with cancer progression in datasets deposited in HCMDB (Figure 1D,E and Figure 2E,F), the malignant phenotypes of *IGF2BP1* were not observed in the EXP00110 dataset and some other datasets in HCMDB. These observations highlight challenges when transferring transcriptomic data to the clinic [38]. The establishment of benchmark standards and assay optimization for analytical and clinical validity must be addressed to improve reproducibility among different studies. Fourth, the findings showing IGF2BP1 overexpression promoted colony formation and chemoresistance (Figure 5) were obtained in two CRC cell lines carrying a *KRAS* G13D mutation according to the Catalogue of Somatic Mutations In Cancer (COSMIC).

IGF2BP1 expression has been linked to therapeutic resistance in several cancers. *IGF2BP1* was a miR-708 target, and overexpression of IGF2BP1 restored cisplatin resistance by promoting Akt phosphorylation in miR-708-overexpressing ovarian cancer cells [39]. In rhabdomyosarcoma, IGF2BP bound to *cIAP1* mRNA directly and promoted the 5′UTR IRES-mediated translation of *cIAP1*, resulting in resistance to *TNFα*-mediated cell death [40]. Inhibition of *IGF2BP1* expression further promoted the treatment efficacy of a BRAF inhibitor and reduced the tumorigenicity of vemurafenib-resistant melanoma cells [41]. To the best of our knowledge, the clinical impact of IGF2BP1 on CRC has not been well addressed. In our findings, IGF2BP1 was associated with advanced stage and was a significant independent marker of poor prognosis; it acted as an oncogene. Moreover, Kaplan-Meier survival analysis using 577 patients in TCGA colorectal datasets showed that high expression of *IGF2BP1* mRNA was linked to poor survival. *IGF2BP1* expression increased in liver metastasized and chemotherapy-resistant CRC patients. The molecular mechanisms underpinning IGF2BP1-initiated therapeutic resistance need further exploration.

## 4. Materials and Methods

### 4.1. Patients

CRC specimens from the Biobank at Taipei Veterans General Hospital were analyzed in our study. The disease stage was assessed based on the American Joint Committee on Cancer (AJCC) staging system, 7th edition. Clinicopathological parameters were determined by searching a computer database. This study followed the guidelines of the Helsinki Declaration and was approved by the ethics committees and institutional review board of the Taipei Veterans General Hospital, Taiwan (Institutional Review Board of the Taipei Veterans General Hospital. IRB number: 2019-04-007CC, 2020-01-010AC) VGHIRB waived the requirement for the use of an informed consent form. This was a retrospective study, and the period in years from the time when patients were diagnosed was not available.

### 4.2. Immunohistochemistry

Immunohistochemistry was performed to examine the expression pattern of *IGF2BP1* protein in colorectal cancer specimens. For tissue microarray, the procedures followed our previous methods and manufacturer’s instructions [42,43]. The antibody used in the study was a polyclonal antibody against human IGF2BP1 (1:100, Ab82968, Abcam, Cambridge, UK). Bound antibodies were visualized using the Envision Detection System (K500711; Dako Denmark A/S), and DAB (diaminobenzidine) was used as a chromogen. The histology score (H-score) is defined as the percentage of IGF2BP1 immunoreactivity region (0 to 100) multiplied byhe intensity of IGF2BP1 expression (0, 1, 2, and 3). A H-score of more than 100 was considered as strong IGF2BP1 immunoreactivity. The assessment of H-scores was performed by a specialist colorectal medical oncologist who was blinded to clinical information.

### 4.3. Bioinformatic Analysis

The TCGA database was used to evaluate the survival impact of *IGF2BP1* expression on CRC (https://www.cancer.gov (accessed on 1 June 2021)). The *IGF2BP1* mRNA expression data were retrieved from the open-access resource (project ID: TCGA-COAD, TCGA-READ) and analyzed by Human Pathology Atlas (https://www.proteinatlas.org (accessed on 1 June 2021)). The HMDCB datasets (EXP00366, EXP00363, EXP00021, EXP00109, and EXP00100) was used to access the expression of *IGF2BP1* under specific pathological conditions (https://hcmdb.i-sanger.com/ (accessed on 1 June 2021)) [44]. The ROC plotter was used to observe the sensitivity and specificity of *IGF2BP1* (probe number: 227377_at*) for classifying chemotherapy responsiveness (http://www.rocplot.org/ (accessed on 1 June 2021)) [45]. COSMIC (https://cancer.sanger.ac.uk/cosmic (accessed on 1 June 2021)) was used to identify mutational status of cell lines.

### 4.4. Cell Lines and Transfection

Two colorectal cancer cell lines, HCT-116 and DLD-1, were used in this study. These cells were initially purchased from ATCC and cultured in RPMI-1640 (Gibco, Waltham, MA, USA) supplemented with 10% fetal bovine serum (FBS, Gibco), penicillin (100 unit/mL, Gibco), and streptomycin (100 μg/mL, Gibco). Cells were maintained at 37 °C in the environment of a humidified incubator with 5% CO_2_. 2 × 10^5^ cells were seeded in 6-well plates overnight, and 3 μg of Myc-DDK-tagged-*IGF2BP1* (RC216226, OriGene, Rockville, MD, USA) or pCMV6-vector control (PS100001, OriGene) were transfected into cells using lipofectamine 2000 transfection reagent (Invitrogen, Waltham, MA, USA) according to the manufacturer’s protocol. The transfectants were selected under a complete RPMI medium with G418 (500 μg/mL, Sigma-Aldrich, Darmstadt, Germany) for 7 days to generate stable cell lines.

### 4.5. Colony Formation Assay

The colony formation assay was performed as follows: 5000 cells were seeded in each well of a 6-well plate with a complete RMPI medium, and the wells were incubated at 37 °C for 7 days. The colonies were stained with 0.5 mL of 0.05% crystal violet (Sigma-Aldrich) for 20 min before colony numbers were determined.

### 4.6. Cell Viability Assay

1 × 10^4^ cells were grown in wells of a 96-well plate in a complete RPMI medium overnight. Then, the medium was removed, and the cells were treated with 5-FU (TTY Biopharm, Taipei, Taiwan) or etoposide (TTY Biopharm, Taipei, Taiwan) for additional 48 h. After the drug treatment, the medium was discarded, and MTT reagent (Sigma-Aldrich) was added to cells for 50 min at 37 °C. The solubilized formazan products were dissolved with DMSO (J.T Baker, Radnor, PA, USA); then, the absorbance was spectrophotometrically quantified with a microplate reader at 560 nm (Spectramax 250, Molecular Devices Corp., San Jose, CA, USA).

### 4.7. Western Blotting

Cells were lysed in RIPA (50 mM Tris HCl, pH 7.4; 150 mM NaCl; 1 mM EDTA; 1% NP 40; 0.1% SDS; 0.5% sodium deoxycholate) buffer with protease inhibitor (Roche, Basel, Switzerland). The protein lysate concentrations were determined using a Pierce BCA Protein Assay Kit (Thermo Fisher Scientific, Waltham, MA, USA) with bovine serum albumin (BSA) as standard. Then, equal amounts of protein samples were separated on SDS-polyacrylamide gels and blotted onto a PVDF membrane (Millipore, Darmstadt, Germany). The following antibodies were used: an anti-Myc tag antibody (1:1000, MA1213161MG, Thermo Fisher Scientific), an anti-Flag M2 antibody (1:3000, F1804, Sigma-Aldrich), an ant-β-actin antibody (1:5000, Ab8226, Abcam, Cambridge, UK), and a chicken anti-mouse IgG-HRP (1:5000, sc-2954, Santa Cruz Biotechnology, Dallas, TX, USA). Immunoblots were visualized with a chemiluminescence detection system (ImageQuant LAS 4000, GE Healthcare Bio-Sciences, Piscataway, NJ, USA).

### 4.8. Statistics

Correlations between clinicopathological variables and immunopositivity in IGF2BP1were analyzed using the χ^2^ test or Fisher’s exact test. Survival was estimated using the Kaplan-Meier plot, and the log-rank test was used to compare survival curves. The *t*-test was used to compare data from the densitometric analysis of foci numbers. The Cox proportional hazards model was applied for univariate and multivariate analysis. A two-sided *p*-value < 0.05 was regarded as statistically significant. SPSS software was used for all statistical analyses.

## 5. Conclusions

IGF2BP1 is a potent oncogenic factor, and enhanced expression of IGF2BP1 was associated with malignant phenotypes in CRC cell lines and poor prognosis in CRC patients. IGF2BP1 could be utilized as a cancer biomarker for disease detection and therapeutic targets for personalized treatment.

## Figures and Tables

**Figure 1 ijms-22-06940-f001:**
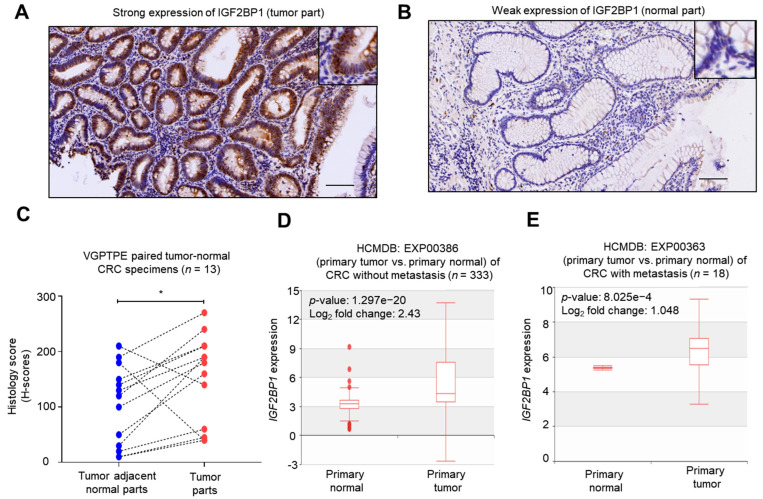
Enhanced expression of IGF2BP1 in colorectal cancer tumor parts compared to normal specimens. (**A**) The representative image of strong expression of IGF2BP1 in a tumor specimen at 100× magnification. Inset: 400× magnification. Scale bar = 100 μm. (**B**) Representative image of the weak expression of IGF2BP1 in the tumor-adjacent normal part at 100× magnification. Inset: 400× magnification. Scale bar = 100 μm. (**C**) Histogram showing the Histology scores (H-scores) in 13 paired tumor-normal specimens. *p*-value = 0.045 by student’s *t*-test. (**D**) Box plot illustrating the expression values of *IGF2BP1* in primary normal specimens and primary tumors without metastasis deposited at EXP00386 in HCMDB. (**E**) Box plot illustrating the expression values of *IGF2BP1* in primary normal specimens and primary tumors with metastasis deposited at EXP00363 in HCMDB. *, *p*-value < 0.05 by student’s *t*-test.

**Figure 2 ijms-22-06940-f002:**
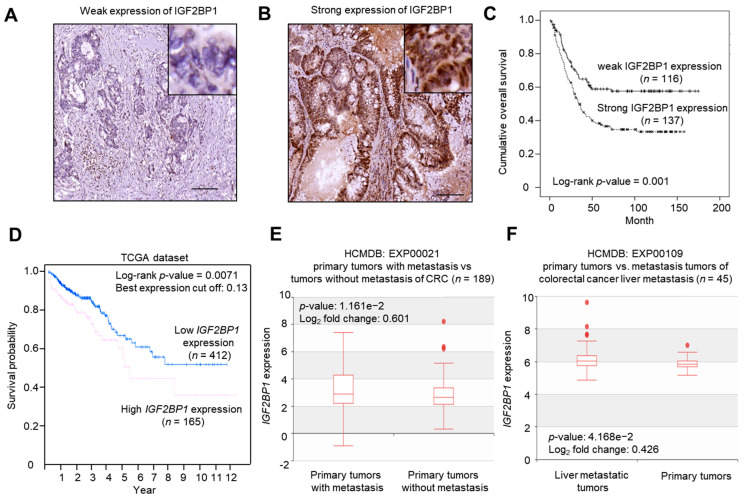
Increased IGF2BP1 expression was associated with advanced CRC progression. (**A**) Representative image showing the weak expression of IGF2BP1 at 100× magnification, Inset: 400× magnification. Scale bar = 100 μm. (**B**) Representative image showing the weak expression of IGF2BP1 at 100× magnification, Inset: 400× magnification. Scale bar = 100 μm. (**C**) Kaplan-Meier survival plot showing the overall survival (OS) of 253 CRC patients with respect to IGF2BP1 expression status. (**D**) Kaplan-Meier survival plot of 577 CRC patients with annotated disease stage and mRNA levels of expression deposited in TCGA datasets. Log-rank *p*-value = 0.0049 under the best expression cutoff. (**E**) Box plot illustrating the expression values of *IGF2BP1* in primary tumors with and without metastasis deposited at EXP00021 in HCMDB. (**F**) Box plot illustrating the expression values of *IGF2BP1* in primary tumors and liver metastasized tumors deposited at EXP00109 in HCMDB.

**Figure 3 ijms-22-06940-f003:**
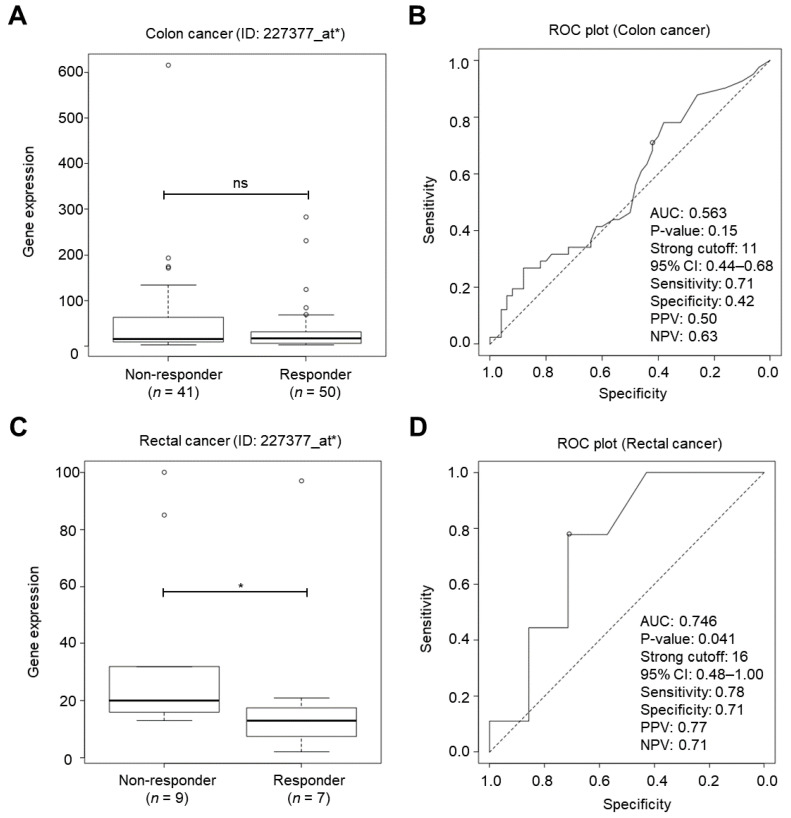
Increased *IGF2BP1* expression was associated with chemoresistance in rectal cancer patients. (**A**) Box plot depicting the expression of *IGF2BP1* (affymetrix probe identify: 227377_at*) in colon cancer patients annotated with chemotherapy responses according to the Response Evaluation Criteria in Solid Tumours (RECIST) criteria. The box plots show the sample maximum (upper end of the whisker), upper quartile (top edge of the box), median (band in the box), lower quartile (bottom edge of the box), and sample minimum (lower end of the whisker) values. ns, non-significance (*p*-value > 0.05 by student’s *t*-test). The outliers were removed for the statistic test. (**B**) Receiver operating characteristic (ROC) plot showing the sensitivity and specificity of *IGF2BP1* (probe: 227377_at*) for classifying chemotherapy responsiveness at the strong cutoff in 91 colon cancer patients. AUC: area under curve. PPV, positive prediction value; NPV, negative prediction value. (**C**) Box plot depicting the expression of *IGF2BP1* (probe: 227377_at*) in rectal cancer patients annotated with chemotherapy responses according to RECIST criteria. The box plots show the sample maximum (upper end of the whisker), upper quartile (top edge of the box), median (band in the box), lower quartile (bottom edge of the box), and sample minimum (lower end of the whisker) values. *p*-value = 0.042 by student’s *t*-test. The outliers were removed for the statistic test. (**D**) ROC plot showing the sensitivity and specificity of *IGF2BP1* (probe: 227377_at*) for classifying chemotherapy responsiveness at the strong cutoff in 16 rectal cancer patients. AUC: area under curve. PPV, positive prediction value; NPV, negative prediction value. *, *p*-value < 0.05 by student’s *t*-test.

**Figure 4 ijms-22-06940-f004:**
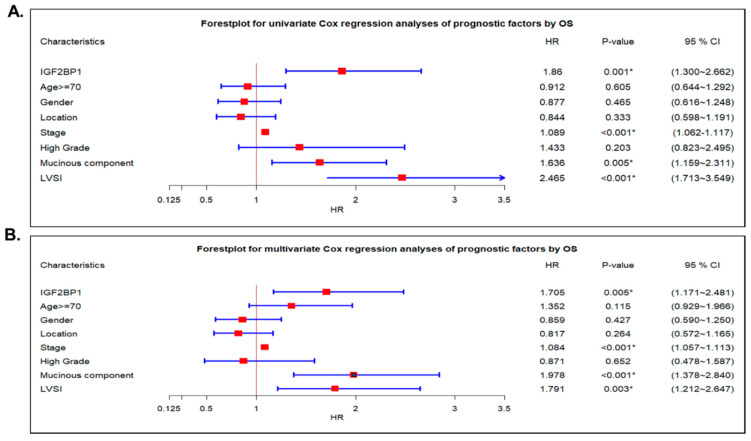
Forest plot of prognostic factors for overall survival according to univariate and multivariate analyses in patients with colorectal cancer. (**A**) In the univariate Cox regression analysis, IGF2BP1, stage, mucinous component and LVSI were significantly prognostic markers. (**B**) In the multivariate Cox regression analysis, IGF2BP1, stage, mucinous component and LVSI were significantly independent prognostic markers. LVSI, lymph-vascular space invasion; HR, hazard ratio. 95% CI: 95% confidence interval. *, significant.

**Figure 5 ijms-22-06940-f005:**
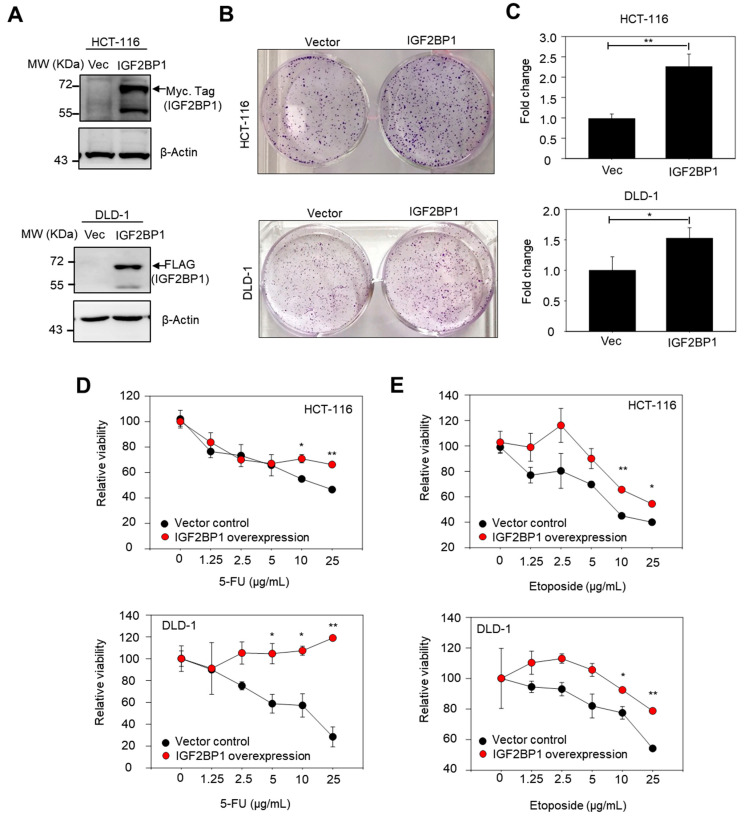
IGF2BP1 overexpression enhanced colony formation efficiency and conferred chemoresistance in HCT-116 and DLD-1 cells. (**A**) Representative western blot analysis confirming the overexpression of IGF2BP1. HCT-116 and DLD-1 cells were transiently transfected for 72 h with IGF2BP1-DDK-Myc. (**B**) Representative images of colonies formed. (**C**) Histograms showed the colony-formation ability of HCT116-and DLD-1-cells overexpressing IGF2BP1. (**D**) Scatter plots illustrating the relative viability of IGF2BP1-overexpressing HCT-116 cells (upper panel) and DLD-1 cells (lower panel) in the presence of 5-fluorouracil (5-FU) for 48 h. (**E**) Scatter plots illustrating the relative viability of IGF2BP1-overexpressing HCT-116 cells (upper panel) and DLD-1 cells (lower panel) in the presence of etoposide for 48 h. Data represented as mean ± SD. *, *p*-value < 0.05; **, *p*-value < 0.01 by student’s *t*-test.

**Table 1 ijms-22-06940-t001:** Association between clinicopathological parameters and IGF2BP1 expression in CRC patients (*n* = 253).

Characteristics		Weak		Strong		*p* Value
		*n*	(%)	*n*	(%)	
Age (y/o)	<70	59	40.4%	87	59.6%	0.043 *
	≥70	57	53.3%	50	46.7%	
Gender	Female	37	41.6%	52	58.4%	0.315
	Male	79	48.2%	85	51.8%	
Pathology	Adenocarcinoma	102	45.5%	122	54.5%	0.553
	Carcinoma	0	0.0%	1	100.0%	
	Mucinous adenocarcinoma	14	51.9%	13	48.1%	
	Signet ring cell	0	0.0%	1	100.0%	
Location	Left	49	42.6%	66	57.4%	0.345
	Right	67	48.6%	71	51.4%	
Stage (AJCC VII)	I/II	51	60.0%	34	40.0%	0.001 *
	III/IV	65	38.7%	103	61.3%	
Grade	Low	106	45.9%	125	54.1%	0.969
	High	10	45.5%	12	54.5%	
Mucinous component	No	70	44.0%	89	56.0%	0.449
	Yes	46	48.9%	48	51.1%	
LVSI	No	93	47.4%	103	52.6%	0.344
	Yes	23	40.4%	34	59.6%	

AJCC, American Joint Committee on Cancer; LVSI, lymph-vascular space invasion; * Significant.

## Data Availability

The data supporting the findings of this study are available from the corresponding author upon request.

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
