# Peer review of "Insulin-Like Growth Factor 2 mRNA-Binding Protein 1 (IGF2BP1) Is a Prognostic Biomarker and Associated with Chemotherapy Responsiveness in Colorectal Cancer"

_ijms, 2021, doi:10.3390/ijms22136940_

Round 1

Reviewer 1 Report

In the manuscript titled, “Insulin-like Growth Factor 2 mRNA-Binding Protein 1 (IGF2BP1) is a Prognostic Biomarker and associated with Chemotherapy Responsiveness in Colorectal Cancer” the authors reported that IGF2BP1 was upregulated in CRC patient tissue samples and has potential to be a novel biomarker for CRC diagnosis and treatment. The authors are recommended to make the following changes.

  1. In the introduction section, authors should discuss briefly about other IGF2BP (IGF2BP2 and IGF2BP3) family of proteins.
  2. In figure 1C, two patients have low IGF2BP1 H-score in tumor tissues compared to adjacent uninvolved tissues. What was the clinicopathological status of these patients? Have these patients received any type of treatment? This should be discussed in the result section
  3. In figure 2, IHC staining score analysis should be performed for figure 2A and 2B.
  4. Figure 3 compared IGF2BP1 expression status in chemotherapeutic responder and non-responder rectal cancer patients. Have the authors checked expression status of IGF2BP1 with the same criteria in colon cancer? The findings should be added in figure 3.
  5. Is figure 4B the analysis of Figure 3C? If yes, figure 3B should be placed after 3C and figure numbering should be changed.
  6. In figure 4D, DLD-1 cells with IGF2BP1 overexpression seem to be resistant to 5-FU treatment. What is the rationale for this? Is there any published literature validating these findings? The authors should include that in the text.
  7. Is the study prospective or retrospective? This should be mentioned in the methods section. In case of prospective study, details of how patients were followed for the entire duration of the study should be included. If the study is retrospective, duration in years when the patients included in the study were diagnosed should be included.
  8. In the methods section, dilution of IGF2BP1 antibody should be added.
  9. Was the histological scoring for IGF2BP1 IHC analysis performed by trained pathologists? It should be mentioned in the methods section. The original article from where the scoring criteria was obtained should be cited.
  10. Conclusion section should be placed after the discussion section and before method sections.

Reviewer 2 Report

The authors of the manuscript „Insulin-like Growth Factor 2 mRNA-Binding Protein 1 2 (IGF2BP1) is a Prognostic Biomarker and associated with Chemotherapy Responsiveness in Colorectal Cancer“ have tried to make a comprehensive analysis of IGF2BP1 expression in CRC. As a source of data, they were using their own IHC data (N=13+253) joined with data deposited at TGCA (N=597) and HCMDB (several datasets). They also used two CRC cell lines (HCT116 and DLD-1) for showing how overexpressed IGF2BP1 influenced colony forming ability and sensitivity to two commonly applied drugs: 5-FU and Etoposide. They also compared seven chemotherapy responders with nine non-responders (Affy probe 227377_at), with respect to the strength of the IGF2BP1 mRNA signal.

First, the authors have shown significantly higher index staining for the expression of IGHBP2 (IHC; N=13) in tumorous tissue as compared with the corresponding non-tumorous tissue in at least 11 paired samples (according to Fig 1A). Still, the authors should better explain the reason for cutoff value when interpreting the H-score. Why was the value of 100 established as a cut off for distinguishing tumors with high and low IGF2BP1 expression? Were there no intermediate expression levels, just high and low?

The images relating to data from HCMDB: 1D, 1E, 2E and 2F, are taken from datasets which support author's original hypothesis. This is best documented, in my view, with respect to dataset EXP00363, where only 3 normal tissues were compared with 15 tumors a total of 18 samples). However, when taking a look at dataset EXP00110, where 12 normal tissues were compared with 18 tumors, the p-value was 0.29. The difference was NOT significant. That was not mentioned, nor was it discussed.

I also think that a prediction for chemotherapy cannot be, nor should it be, based on the strength of the signal obtained with one Affy probe and a total of 16 samples. That is an important part of the study – it  was supposed to be  predictive : associating the marker with the response to therapy. The ROC analysis should have included 95% confidence interval, sensitivity, specificity, positive and negative predictive value.

I wonder why the authors did not stratify their patients and present their original data/results.

With respect to Figure 2D (TGCA dataset) which can be easily found here (https://www.proteinatlas.org/ENSG00000159217-IGF2BP1/pathology/colorectal+cancer), the Human Protein Atlas clearly states that IGF2BP1 is not prognostic in colorectal cancer. Still, the p-value was shown to be 0.0049 (correctly shown on Figure 2D). This raises serious issues relating to statistical analysis:

With respect to Table 2: AJCC criteria for establishing a new prognostic marker clearly state that only  a multivariate approach is relevant to this type of research. This means that multivariate analysis must be performed with all variables included in an univariate Cox regression. Forest plot of HR is highly desirable. An even better approach would be a tree structured survival analysis.

With respect to Figure 1: ROC analysis is required with a full description; 95% confidence interval, sensitivity, specificity, positive and negative predictive vale. Also, it should be presented  after a Cox multivariate regression. The same applies to Figure 2: it should be presented after a Cox multivariate regression.

With respect to Table 1: data should be presented according to the AJCC Cancer Staging Manual 8th edition. Also, how were the p-values calculated?

In conclusion, the authors have an impressive number of clinical samples which, if properly stratified, have a great potential to answer some important questions relating to IGF2BP1 functioning in colorectal cancer, in a specific population. Instead, the authors have presented several images, and the majority of data,  drawn from specific HCMDB datasets, which, as explained earlier (EXP00363 vs EXP00110), support their original hypothesis. As a result, with respect to that and everything I already stated, my impression is that the content in this manuscript tends to be highly biased.

Thank you.

Round 2

Reviewer 1 Report

In the updated version of the manuscript titled, “Insulin-like Growth Factor 2 mRNA-Binding Protein 1 (IGF2BP1) is a Prognostic Biomarker and associated with Chemotherapy Responsiveness in Colorectal Cancer” the authors have taken into accounts the reviewers' comments and incorporated the recommended corrections. The review would be interesting to readers in cancer biology.

Reviewer 2 Report

I am satisfied with the improved version of the manuscript "Insulin-like Growth Factor 2 mRNA-Binding Protein 1  (IGF2BP1) is a Prognostic Biomarker and associated with  Chemotherapy Responsiveness in Colorectal Cancer". The authors significantly improved this manuscript according to suggestions given and questions asked.

There are still two minor objections:

The subtitle 2.2. needs to be changed: „IGF2BP1 was Associated with Malignant Phenotypes in CRC Patients“. The content of the subchapter clearly shows aggressive (clinical) behaviour (and not malignant phenotype) of the CRCs with respect to IGF2BP1“.

With respect to limitations of this study, please state the mutational status of the two colon cancer cell lines, according to COSMIC (for example, both cell lines have affected codon 13 of the K-RAS oncogene).

Thank you.
